# DNA-Based Mechanical Sensors for Cell Applications

**Xiaoya Sun, Pengyan Hao and Na Wu ***

The Key Laboratory of Biomedical Information, Engineering of Ministry of Education, Institute of Analytical Chemistry and Instrument for Life Science, School of Life Science and Technology, Xi'an Jiaotong University, Xianning West Road, Xi'an 710049, China

**\*** Correspondence: wuna2017@xjtu.edu.cn

**Abstract:** Cells constantly experience mechanical forces during growth and development. Increasing evidence suggests that mechanical forces can regulate cellular processes such as proliferation, migration, and differentiation. Therefore, developing new tools to measure and manipulate cellular mechanical forces is essential. DNA nanostructures, due to their simple design and high programmability, have been utilized to create various mechanical sensors and have become a key tool for studying mechanical information in both cellular and non-cellular systems. In this article, we review the development of DNA-based mechanical sensors and their applications in measuring mechanical forces in the extracellular matrix and cell–cell interactions and summarize the latest advances in monitoring and manipulating cellular morphology and function. We hope that this review can provide insights for the development of new mechanical nanodevices.

**Keywords:** DNA nanostructures; mechanical forces; cell

## 1. Introduction

In recent years, there has been increasing awareness of the importance of cellular-generated mechanical forces in biological systems. Each cell possesses mechanical forces that regulate biological functions or information exchange. These forces can not only originate from within the cell but can also be externally applied [1]. The main source of force generated within cells is derived from the cytoskeleton [2]. These forces play crucial roles in cellular shape changes, movement, and tissue function. External forces primarily arise from neighboring cells or the extracellular matrix (ECM) and can be mediated and transmitted through transmembrane adhesion protein receptors, such as integrins and cadherins [3]. These forces are essential for cellular adhesion, proliferation, differentiation, and other processes. Moreover, some studies have demonstrated a close correlation between changes in the mechanical forces generated by cells or their responses to mechanical stimuli and human diseases. For example, integrin-mediated mechanical strain on mesangial cells in renal hypertension leads to glomerulosclerosis [4]. Mechanical force damage can cause a range of different manifestations of heart disease [5]. Cancer cell metastasis is also closely associated with changes in the mechanical forces of the tumor microenvironment [6]. Therefore, investigating the mechanical forces within cells has profound implications for understanding biological functions and for the diagnosis and treatment of diseases.

Currently, there are numerous techniques available for studying cellular mechanics. For example, the single-molecule force spectroscopy method, atomic force microscopy (AFM) [7], can be used to trace force curves by bending a cantilever; optical tweezers [8] and magnetic tweezers [9] directly apply forces to cells using light or a magnetic field to obtain mechanical information. Although these sophisticated instruments offer detection limits in the pN range and high resolution, direct manipulation of the sample may damage it, and there are limitations to single-cell testing throughput [10]. In addition to the aforementioned techniques, traction force microscopy (TFM) and micropost array detectors [11] can be used to map mechanical force information by measuring substrate deformation,

but low resolution and extensive computational processing are still issues [12]. Therefore, developing new techniques for detecting cellular mechanics is imperative.

In 2011, Stabley et al. [13] developed a fluorescence-based molecular tension sensor by combining single-molecule force spectroscopy with traction force microscopy. This sensor used the contraction and stretching of polyethylene glycol (PEG) polymer probes to produce a fluorescent signal that reflects the force transmitted by a single receptor in a cell. Later, the use of nucleic acids [14], peptides [15], or proteins [16] as probes for mechanical measurements was developed. Among these probes, due to its unique properties, DNA stands out.

As we all know, DNA has gradually become a powerful material in nanotechnology. The specific base-pairing rules of DNA allow it to be used as a building block to design nanoscale structures of desired shapes and sizes. These structures can be utilized as molecular machines to perform complex tasks. For example, switchable nanostructures [17] can be constructed that change their structure in response to external stimuli such as light, temperature, pH, and small molecules. Such stimulus-responsive platforms can be widely used in biosensing, bioimaging, and biotreatment. In addition to these traditional chemically responsive nanodevices, mechanically sensitive DNA nanostructures, which utilize their mechanical properties to sense changes in force and undergo structural changes, have attracted increasing attention as a means to measure mechanical information in cells or non-cellular systems through rational design. Therefore, due to its simple design, precise control, high programmability, and biocompatibility [18], DNA has gradually become a powerful tool for studying cellular mechanics.

In this review, we first introduce the development of DNA as a mechanical sensor and review the latest research progress in the application of DNA-based mechanical sensors in cells, including measuring forces between cells and the ECM, forces between cells, and the monitoring and regulation of cellular mechanical function. We hope this review will provide some assistance for the future development of DNA-based mechanical sensors.

## 2. DNA-Based Tension Probes

Molecular recognition (base-pairing) of nucleic acids has been shown to enable the construction of various nanostructures [19] whose mechanical properties can be used to measure the strength of interactions with target molecules. To better understand the mechanical properties of nucleic acid nanostructures, previous studies have combined single-molecule force spectroscopy (SMFS) with computational modeling to directly obtain their mechanical properties [20]. By anchoring one end of the target nucleic acid to a substrate and attaching the other end to a force sensor, a specific rupture force of the nucleic acid can be measured by applying a certain amount of stretching force. These basic mechanical properties of DNA structures provide theoretical support for constructing DNA-based tension probes.

In 2003, Albrecht et al. [21] first proposed using the force required to break DNA double-stranded molecular bonds as a tool for single-molecule force measurement. This tool was designed as a long oligonucleotide chain with fluorescent groups, conjugated with two short oligonucleotides at both ends, which were modified with PEG to attach to a soft polydimethylsiloxane (PDMS) and activated glass surface, respectively. Force was applied at one end, and when the DNA duplex was broken, the mechanical information was reflected by the fluorescent signal (Figure 1a). Through this design, the research group demonstrated the specificity and nonspecific binding of antigen-antibody interactions in the field of protein arrays. Additionally, due to the different strengths of interactions between different base pairs, by changing the type and number of bases, different threshold forces can be designed. Wan et al. [22] modified the biotin and integrin ligand RGDfk at different positions of the DNA double helix to provide different rupture force anchoring points and integrin binding sites. They used this method to prepare DNA duplexes with different numbers of base pairs, designing a series of DNA probes (12, 16, 23, 33, 43, 50, 54, and 56 pN) with different tension thresholds. They utilized tension sensors with

varying thresholds to analyze the mechanical information between the B-cell receptor (BCR) and antigen, providing evidence that the activation of B-cell receptor IgM-BCR is mechanistically dependent, and memory IgG-BCR or IgE-BCR can be activated even at forces lower than 12 pN or without mechanical force, revealing the mechanical sensitivity of BCR in antigen sensing. Therefore, the broad range of interaction forces among molecules can be detected by rationally designing DNA double-stranded structures.

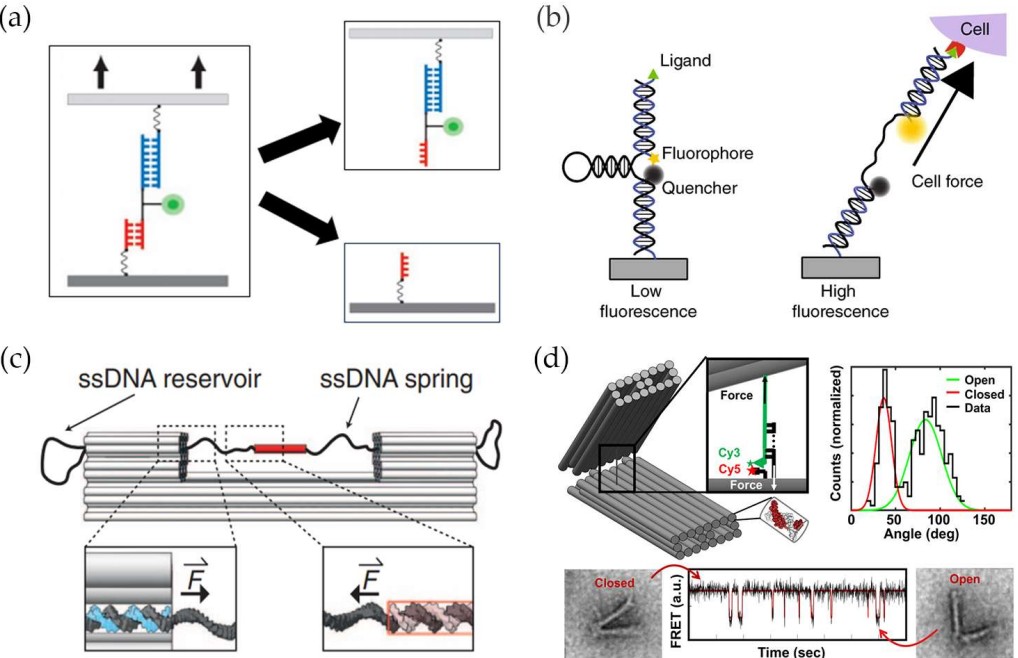

**Figure 1.** DNA-based tension probes: (**a**) The differential force test compares the breaking force of a sample bond (red) to a known reference bond (blue) [18]. Copyright 2003, American Association for the Advancement of Science. (**b**) Principle of measuring force direction and magnitude using molecular force microscopy (MFM) [21]. Copyright 2017, Springer Nature. (**c**) Structure of a DNA origami force clamp, where ssDNA links the target system (red rectangle) and the force applied to the target system (dsDNA) is detected based on the ssDNA shearing conformational change [23]. Copyright 2016, American Association for the Advancement of Science. (**d**) Schematic diagram of a nanoscale DNA-based force spectrometer (nDFS), where the device closing and opening is reflected by transmission electron microscopy (TEM) and Förster resonance energy transfer (FRET) results [24]. Copyright 2022, American Chemical Society.

Other than duplexes, due to the mechanical information conversion that occurs during DNA hairpin folding and unfolding, DNA hairpins have also been used as popular building blocks for tension probes. Zhang et al. [14] designed a novel molecular tension probe, in which a hairpin serves as a "switch", containing a quenched fluorophore-modified strand anchored to the base and a fluorescently labeled strand bearing an RGD-adhesive peptide that interacts with integrin. Upon reaching the tension threshold, the hairpin unfolds and releases a fluorescent signal, visualizing the tension of integrin. Su et al. [23] developed a light-responsive polymer force clamp (PFC), which is centered on a gold nanorod as a photothermal converter and coated with a thermoresponsive poly-N-isopropylmethacrylamide) (pNIPMAm) shell. The DNA hairpin anchored to the gold particle is coupled to the PFC by click chemistry. Under near-infrared (NIR) illumination, the PFC collapses, and the hairpin unfolds, converting mechanical signals into optical signals, providing a method for mechanical manipulation at the nanoscale time scale. Additionally, Brockman et al. [25] introduced a molecular force microscope (MFM), which combined fluorescence polarization microscopy and fluorescently labeled molecular tension probes. When the receptor force exceeds the tension threshold, the hairpin opens, and the anchored probe determines the

size and direction of the cell force as DNA and fluorescent groups rotate in the direction of the receptor force (Figure 1b). This technique has been used in integrin-mediated mechanical information in platelets and fibroblasts. Furthermore, studies have shown [24] that different kinetic and thermodynamic folding and unfolding behaviors of DNA hairpins can be induced by adjusting the stem/loop length and the stem GC contents, thereby constructing probes with different tension thresholds. Compared to DNA duplexes, DNA hairpins have a simpler structure, composed of only a single oligonucleotide, and a higher folding efficiency. Meanwhile, the molecular interactions can be regulated according to the dynamic folding and unfolding of the hairpin, and the accuracy can be improved by repeated measurements.

In addition to the simple DNA duplex and hairpin structures, DNA origami structures have precise nanoscale arrangements and can be used to modify target molecules at specific sites for research purposes. Moreover, compared to the above-mentioned two DNA structures, DNA origami structures have greater rigidity and better stability, making them a useful tool for mechanical sensing. Nickels et al. [26] developed a DNA origami-based nanomechanical force clamp, in which a M13mp18 scaffold with multiple cloning sites was fixed in the middle of a spring, and single-stranded DNA was anchored to fixed anchor points at both ends. By utilizing the entropy-driven behavior of single-stranded DNA, a certain force can be applied to the system, while different lengths of single-stranded DNA can be adjusted to provide different forces (Figure 1c). Mechanical monitoring based on the FRET effect between the donor and acceptor on the two arms has demonstrated the mechanical information between two Holliday junction (HJ) conformations and the mechanical force induced by TATA-binding protein (TBP) when bending the DNA duplex. Additionally, Wang et al. [27] designed a nanoscale DNA force spectrometer (nDFS) based on dynamically adjustable DNA origami hinges, which can be turned to a more open or closed state by adding or replacing some scaffold chains to apply stretching or compression forces. This structure has demonstrated compression forces sufficient to induce the deformation of a DNA duplex and the ability to unfold nucleosomes with smaller angles by adjusting the tension. Based on this, the research group [28] developed a nanoscale force spectrometer based on nanocalipers (Figure 1d), which was designed with different hinge angles connected by DNA interactions at the hinge vertex and is capable of applying forces of tens of piconewtons at the nanoscale based on the thermal fluctuation dynamics of nucleotides, providing support for the development of new force spectroscopy techniques.

In summary, DNA-based tension probes, whether duplex, hairpin, origami, or other non-canonical nucleic acid structures, such as G-quadruplex and i-motif, can perceive mechanical forces and induce conformational changes compared to traditional force measurement techniques. They typically reflect mechanical information through fluorescence signals. These probes have the advantages of simple operation, high sensitivity, and high throughput, and have gradually become an ideal tool for studying mechanical transduction. Therefore, the application of these DNA tension probes can be extended to the field of biophysics. Next, we will provide a detailed introduction to the latest research progress using this tool in cellular mechanical information.

## 3. Force Measurement at Cell–Extracellular Matrix

The extracellular matrix (ECM) is a highly dynamic structure that continuously reshapes itself through mechanical interactions with transmembrane adhesive receptors on cells to maintain homeostasis during the growth and development of cells [29]. Through signal transduction mediated by adhesive interactions, the ECM is closely related to the communication [30], migration [31], tissue repair [32], and other processes of cells. The composition and stiffness of the ECM are related to many diseases, such as cancer occurrence and metastasis [33]. Therefore, studying the forces between cells and the ECM has profound significance.

In 2013, Wang et al. [34] developed a method called the DNA tension gauge tether (TGT), where one strand is fixed to a surface while the other is modified with a ligand that interacts with a receptor on the cell surface. By changing the DNA base and length, a series of TGTs with different tension thresholds can be designed. When the force generated by receptor activation exceeds the tolerance of the TGT, the double strand breaks, producing a fluorescent signal that reflects the force generated by the receptor–ligand interaction. This TGT was designed to study the mechanical force between integrin and the cyclic RGDfK peptide ligand, indicating that membrane tension plays a dominant role during the early stage of adhesion. It also showed that the molecular tension required for Notch receptor activation is less than 12 pN, or even no force at all. However, due to mechanical events and cell size, the signal from a single fluorescent group is difficult to quantify and requires high-resolution microscopy [35]. Therefore, to further improve the accuracy and sensitivity of TGT measurements, Ma et al. [36], inspired by PCR, proposed the mechanically induced catalytic amplification reaction (MCR), which directly visualized the amplification product using rolling circle amplification (RCA) and fluorescence in situ hybridization and measures the signal on a high-throughput ELISA reader, achieving a hundredfold signal enhancement. They quantified the mechanical force mediated by integrin and demonstrated the ability of drug screening, indicating drug dose-dependent effects of integrin tension. Duan et al. [37], based on nucleic acid hybridization chain reaction (HCR), exposed the anchored chain on the base when the mechanical force reached the threshold of duplex breakage, initiating the HCR reaction to amplify the mechanical signal. The amplified signal can be directly imaged by fluorescence microscopy or read out by an ELISA reader (Figure 2a). Compared to traditional TGT, this method improved the signal-to-noise ratio by an order of magnitude. This device can be used to screen the effect of the mechanical force of different cells on drug treatments and the effect of different drugs on cell mechanical signals. Using platelets as an example, they demonstrated that the three publicly available anticoagulant drugs have a similar IC50 (half-maximal inhibitory concentration). Therefore, this device can provide a basis for rapid and efficient drug screening.

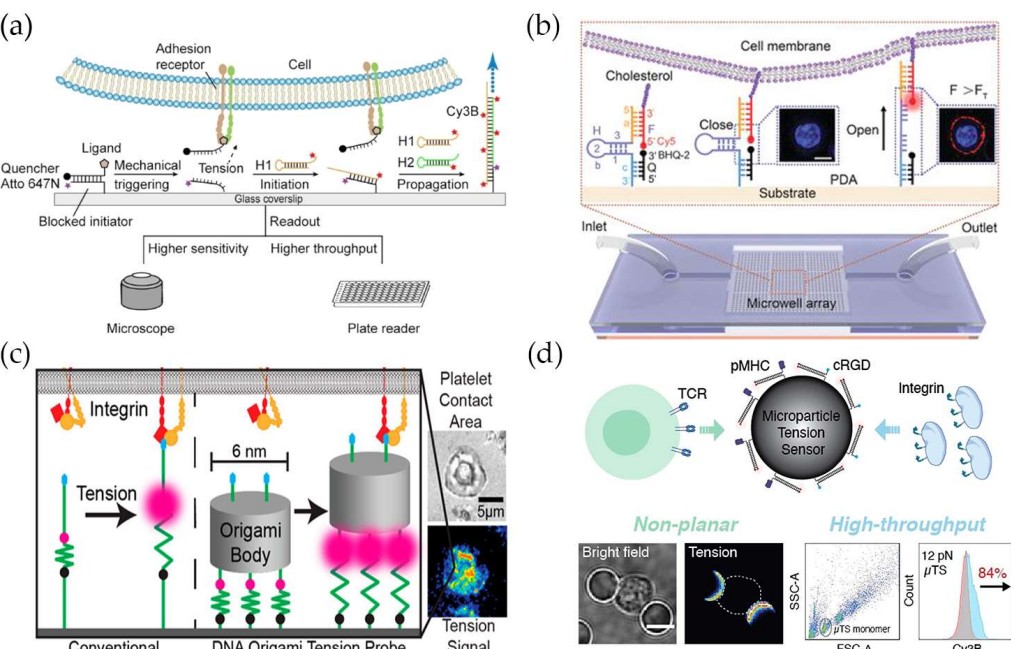

**Figure 2.** DNA nanodevices for force measurement of cells and extracellular matrix: (**a**) Schematic representation of DNA-based hybridization chain reaction (HCR) for mechanical signal amplification, applicable for drug discovery, screening, and identification of cellular mechanical phenotypes [34].

Copyright 2021, Wiley Publishers. (**b**) Structural diagram of a DNA tension gauge platform based on microfluidic chips, enabling high-throughput parallel mechanical force measurement of multiple cells [35]. Copyright 2022, Wiley Publishers. (**c**) Design schematic of a DNA origami tension probe (DOTP) with three parallel probes, suitable for studying the interaction forces between multiple receptor ligands [38]. Copyright 2018, American Chemical Society. (**d**) DNA-based microparticle tension sensors (μTS), anchoring ligand-modified tension probes to silica particles to investigate non-planar mechanical transduction [39]. Copyright 2021, Wiley Publishers.

In addition, although the methods mentioned above have utilized RCA or HCR to enhance mechanical signals, these methods still suffer from low throughput and cannot simultaneously detect multiple cells. Therefore, in order to improve cell throughput and resolution, Hang et al. [40] established a high-throughput DNA tension gauge platform based on microfluidic chips, where the fluorescent signal from DNA hairpin serves as an indicator of cell force, allowing for tens of thousands of cells per chip and pN-level resolution (Figure 2b). It has been demonstrated that drug-resistant cells in tumor cells exhibit stronger mechanical forces than drug-sensitive cells, and there are mechanical differences between cells in tumor tissues and those in pleural effusions, laying the foundation for understanding the mechanism of tumor metastasis. Moreover, due to the complexity of the cellular microenvironment, there may be a possibility of nucleases degrading DNA probes. Zhao et al. [38] developed a tension probe based on peptide nucleic acid (PNA), which was composed of peptide N-(2-aminoethyl)-glycine and possesses the ability to resist DNA nuclease degradation. Designing DNA/PNA probes significantly improved stability and maintained the ability to measure cell force. Recently, Pawlak et al. [41] developed a "rupture and delivery" tension gauge tether (RAD-TGT) that utilizes flow cytometry to detect mechanical signals from thousands of cells surrounding a broken double-stranded DNA probe. And designing DNA–protein interactions allows for the recording of events when the broken oligonucleotide enters the cell, improving early TGT fluorescence chain internalization by cells [42].

Of course, some of the mechanical transduction in cells is not simply a single receptor–ligand interaction, but rather involves the oligomerization of multiple receptors on the membrane. Therefore, DNA origami can be used as building blocks to precisely arrange the number and spacing of ligands and explore the mechanical information borne by multiple receptors. Dutta et al. [39] first used DNA origami to study cellular mechanics and developed a DNA origami tension probe (DOTP) where each origami contains three hairpins organized in parallel to bind multiple ligand receptors (Figure 2c), providing support for achieving multivalent interactions. Using this probe, they were able to map the mechanical information of human platelet–integrin during adhesion and activation, with forces in individual platelets showing high heterogeneity. This represents an important step towards studying multiple ligand binding sites in cells. The aforementioned methods described are based on research on planar substrates, but there are still some physiological activities that occur on non-planar geometries in vivo, such as phagocytosis [43] and the formation of immune synapses [44]. Hu et al. [45] developed a DNA-based microparticle tension sensor (μTS) fixed on cell-sized magnetic beads, which allowed for mechanical analysis using fast readouts and high-throughput flow cytometry (Figure 2d). This method directly visualized the mechanical forces between TCR-pMHC in immunological synapse formation and the mechanical pharmacology of platelet adhesion and aggregation. Although this method has some limitations, such as the relative hardness of silica particles compared to cells that may affect signal intensity, the advantages of high throughput and high spatial resolution still provide a method for further studying mechanical transduction on non-planar interfaces.

## 4. Force Measurement at Cell–Cell Junctions

The above-mentioned research primarily focuses on the interaction between cells and the ECM, while the mechanical information between cells also plays a significant role in the growth and development of organisms. For example, the tension generated by cell–cell interactions not only acts as a barrier during embryonic morphogenesis and

in stationary adult tissues [46] but also controls their proliferation [47]. Additionally, mechanical forces between cells can serve as a tool for intercellular communication, rapidly converting mechanical signals into signals for cell communication through cell–cell contact or some secreted factors [48]. Therefore, the development of new methods for measuring intercellular forces is of great significance in biomedical research.

### 4.1. Measuring Intercellular Forces at the Single-Cell Level

The mechanical forces between adjacent cells are primarily generated through receptor–ligand interactions on the cell membrane surface [49]. To reflect the mechanical forces experienced by these receptor–ligands at cell–cell junctions, Ma et al. [50] developed a gold nanoparticle-based solid-supported lipid bilayer (SLB) tension probe, which simulates the cytoplasmic membrane. They used fluorescence signals generated by DNA hairpin breakage to measure the forces generated by receptor–ligand interactions during intercellular signal transduction (Figure 3a), and used this method to measure the mechanical information of immune receptors in T and B cells. However, on fixed SLBs, most of the forces detected by the probe come from the vertical direction, while the forces generated parallel to the membrane are limited [51]. Therefore, to measure intercellular forces more accurately, Zhao et al. [52] reported a novel membrane DNA tension probe (MDTP), which used cholesterol at one end anchored to the membrane and interacting with a receptor–ligand of another cell through transmembrane adhesion proteins (integrins, E-cadherin, etc.) at the other end. The mechanical forces mediated by cell–cell connections can be visualized through fluorescence signals (Figure 3b). The study demonstrated the intercellular tension mediated by integrin and E-cadherin. The probe has the advantages of simple operation and high-sensitivity imaging, but DNA hairpins can only be opened within a small threshold range [24], making it challenging to measure a wide range of mechanical information. Additionally, the anchoring persistence of the DNA probe on the membrane is also an issue. Building upon this work, Zhao et al. [53] developed a second-generation DNA-based membrane tension ratiometric probe (DNAMeter), which contains two different tension thresholds of DNA hairpins. The fluorescence signals from two orthogonal fluorescent and quenched groups respond to low, medium, and high intercellular forces, greatly expanding the range of force measurement (Figure 3c).

### 4.2. Measuring Intercellular Forces at the Collective-Cell Level

The biological processes of organisms are complex and diverse and not only involve the mechanical forces between individual cells, but also the collective cell mechanical behaviors and functions are of significant importance. For example, in processes such as tissue growth, embryonic morphogenesis, and wound healing, there are many collective cellular behaviors that are usually regulated by mechanical forces mediated by E-cadherin [54]. The second-generation tension probe (DNAMeter) [53] described in the previous section quantifies the tension between cells during collective cell migration. Using a polydimethylsiloxane (PDMS) substrate and a continuous monolayer of MDCK cells to simulate the wound-healing process, the initial introduction of the DNAMeter resulted in minimal intercellular forces. However, after 12 h of cell migration and growth, adding a fresh DNAMeter revealed that intercellular tension increased linearly with the distance from the leading edge of migration. Wang et al. [55] used molecular tension fluorescence microscopy (MTFM) to quantify the mechanical forces mediated by integrin during the collective migration of epithelial cells in wound healing, using the fluorescence signal released by DNA spring fracture (Figure 4a). They found that there was a high force present at the wound edge, which was spatially correlated with the energy consumed. Recently, this research group further studied the mechanics and energy cost of collective cell migration in confined microchannels [56]. The results showed that the smaller the confined space, the higher the mechanical force and energy cost, and the cells tended to migrate in an orderly manner (Figure 4b). Therefore, due to the simple design and operation of this probe, it

is advantageous for us to understand various collective cellular behaviors and functions within the body.

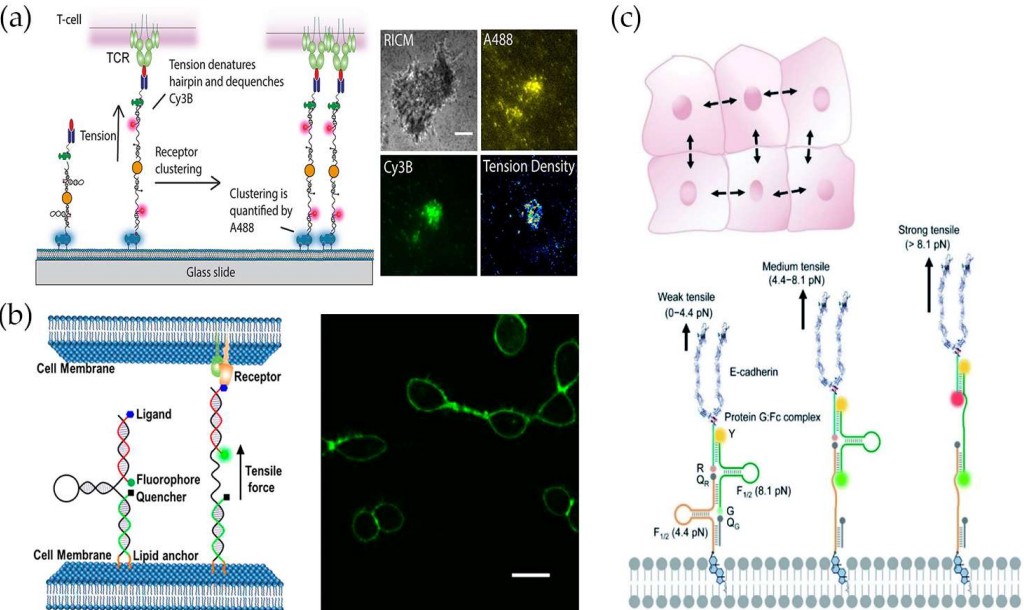

**Figure 3.** DNA tension sensors for force measurement at single-cell level: (**a**) Schematic diagram of a gold nanoparticle-based proportional tension probe, using a solid-supported lipid bilayer (SLB) to measure the interaction forces between adjacent cells [47]. Copyright 2016, American Chemical Society. (**b**) Design diagram of a membrane DNA tension probe (MDTP), which directly measures the mechanical information between adjacent living cells through hydrophobic interactions with cholesterol [49]. Copyright 2017, American Chemical Society. (**c**) Schematic diagram of the second-generation DNA-based membrane tension ratiometric probe (DNAMeter), designed with DNA hairpins of different tension thresholds to measure forces within different ranges of low, medium, and high tension [50]. Copyright 2020, Royal Society of Chemistry.

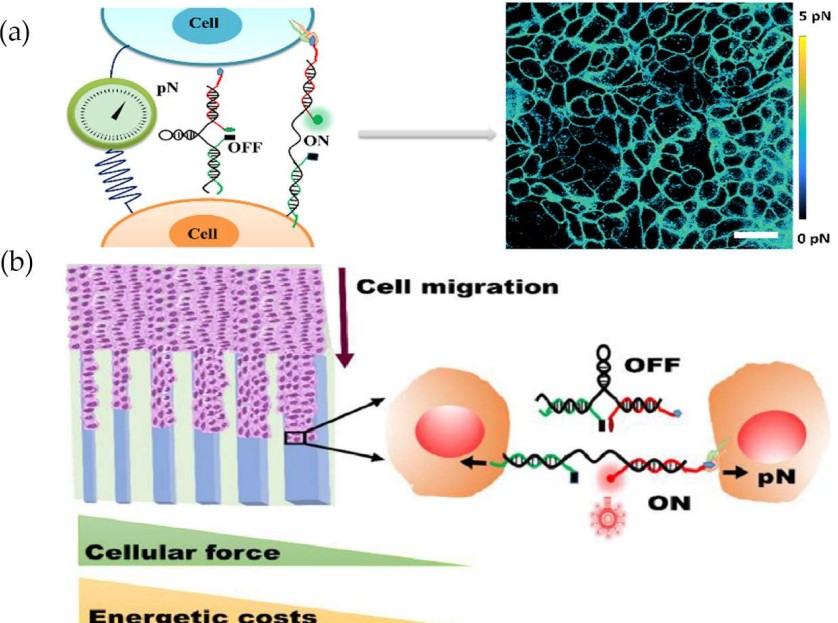

**Figure 4.** DNA tension sensors for force measurement between collective-cell levels: (**a**) Principle of measuring intercellular forces during wound healing in epithelial tissue, depicting a representative

mechanical map using fluorescence signals [52]. Copyright 2020, American Chemical Society. (**b**) Schematic diagram of cell migration and force measurement in confined microchannels, in which the probe switches from "OFF" to "ON" under the influence of cellular forces [53]. Copyright 2022, The Company of Biologists.

## 5. Monitoring and Regulating Cellular Mechanical Functions

The above content summarizes the use of DNA-based mechanical sensors for measuring forces in cell–ECM and cell–cell interactions. DNA-based structures, such as a DNA duplex, hairpin, or spring, can be used to directly visualize and study cellular mechanical forces. In addition, using DNA mechanical probes, it is possible to directly monitor intracellular mechanical information, such as exploring the mechanisms related to mechanics in cell contraction, maturation, and apoptosis. Furthermore, due to the programmability of DNA nanostructures [18], it is possible to regulate cellular function directly or indirectly through various stimuli. The following will describe how DNA tension probes can be used to monitor and regulate cellular mechanical function.

Many cells respond to endogenous or exogenous stimuli with mechanical responses. For example, smooth muscle cells [57] can contract under stimulation mediated by actin and myosin, regulating physiological functions, such as intestinal peristalsis and bronchial movement. However, excessive cell contraction can lead to airway narrowing and cause asthma [58]. Jo et al. [59] used DNA tension probes to study the effect of neurotransmitters (e.g., histamine) on smooth muscle cell contraction (Figure 5a). The results showed that enhancing focal adhesions (FAs) prolonged the time of cell contraction and required higher forces to break the tension probe. Therefore, studying the regulation of FA stability could provide some help in treating diseases such as asthma, characterized by airway narrowing. It is known that the rhythmic beating of the heart during development depends on mature cardiac muscle cells (CMCs) [60]. Rashid et al. [61] used DNA tension probes with different threshold values to study how mechanical forces control the maturation of CMCs (Figure 5b). The study found that when DNA adhesion tethers with greater force tolerance were applied, CMCs exhibited a mature state, such as cell contraction, elongation, increased calcium ion production, and increased expression of proteins associated with maturation. Therefore, regulating the maturation of CMCs through mechanical forces may greatly help in the future treatment of heart disease.

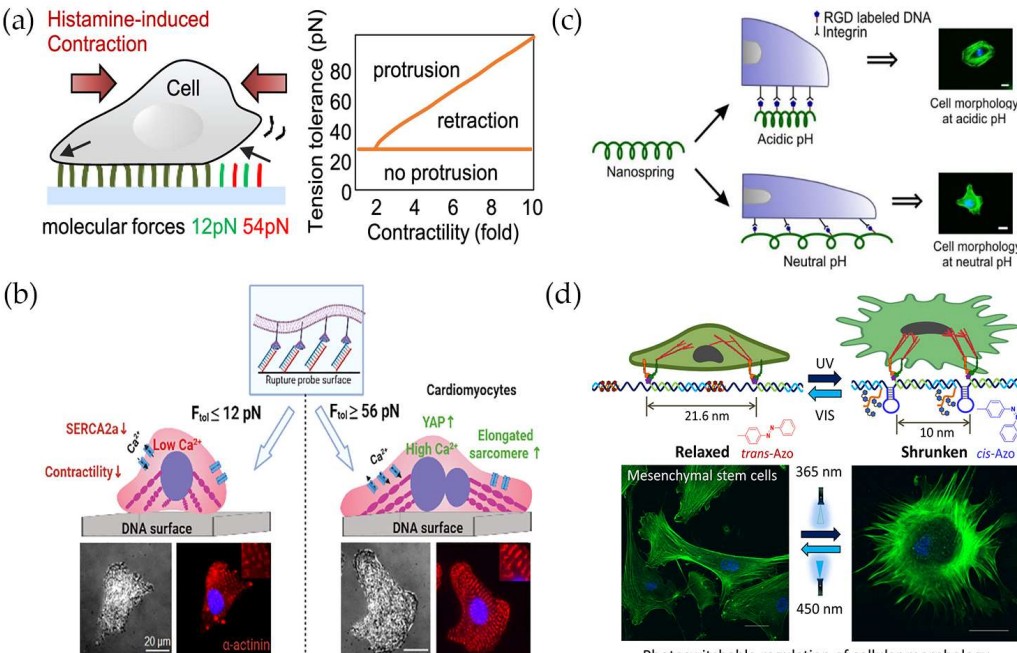

**Figure 5.** DNA-based mechanical sensors for monitoring and regulating cellular mechanical functions: (**a**) Used to explore the force transmission mechanism during histamine-induced smooth muscle cell

contraction process, monitored by real-time fluorescence imaging and molecular measurements [56]. Copyright 2021, American Chemical Society. (**b**) Utilizing DNA adhesion probes with specific tension thresholds to study the impact of forces on cardiac muscle cell (CMC) maturation, monitoring changes in cell morphology and protein expression, etc. [58]. Copyright 2022, American Chemical Society. (**c**) DNA origami-based nano-springs containing i-motif structures, using pH-adjustable nano-springs to control cell morphology and function. At acidic pH, the i-motif deforms, causing the spring to contract and altering cell morphology. [61]. Copyright 2021, American Chemical Society. (**d**) DNA polymers that can be optically switched on and off based on azobenzene, regulating cell morphology through the cis–trans isomerization of azobenzene molecule [62]. Copyright 2021, Wiley Publishers.

Not only can DNA tension probes be used to directly monitor mechanical information such as cell contraction and maturation, providing a basis for subsequent regulation of cell function, but stimuli such as light, pH, temperature, and DNA strand displacement can also be used to directly regulate cell morphology and function. For example, Zhang et al. [63] designed a nano-spring containing multiple hairpins that dynamically regulate the aggregation or separation of integrin receptors on the cell surface through DNA strand displacement, by modifying RGD to interact with integrins. They found that polymeric RGD significantly enhanced cell binding affinity, adhesion, and diffusion. Upon adding cDNA2, the cell morphology changed from numerous pseudopodia to a smooth surface, further regulating cell function. Based on this, the group developed a pH-driven interlocked DNA nano-spring [62], which induced spring contraction in the acidic tumor microenvironment, causing T cell receptors CD3 and CD8 to aggregate, activating T cells and enhancing tumor immunity. Karna et al. [64] prepared a DNA origami-based nano-spring with an i-motif (a pH-responsive motif) embedded in the spring coil. When the pH changed from neutral to acidic, the i-motif folded from a single strand to a quadruplex, causing the nano-spring to contract and receptors to aggregate, leading to changes in cell shape and movement (Figure 5c), such as inhibiting Hela cell migration in an acidic matrix, which is expected to be beneficial for inhibiting tumor cell metastasis. Li et al. [65] developed a reversible shearing DNA-based tension probe (RSDTP) that can measure a wide range of forces (4–60 pN) transmitted by cells, and modified the probe's ring with light-cleavable groups. The probe forms an irreversible state under UV light, thereby changing cell morphology. Sethi et al. [66] created a photo-switchable DNA nanostructure containing azobenzene (Figure 5d), which dynamically regulates the distance between cell adhesion peptides (RGD) through the conformational changes of azobenzene under UV and visible light, thus regulating cell morphology.

In summary, DNA-based mechanical sensors can not only measure and regulate mechanical information such as cell contraction and maturation but can also further regulate cell morphology and function by adjusting receptor aggregation or separation. They play an important role in the growth, development, diagnosis, and treatment of diseases in living organisms.

## 6. Summary

In this review, we first introduced several forms and developmental processes of DNA-based mechanical sensors, including DNA duplex, hairpin, origami, and other structures, providing technical support for the development of more effective probes. Next, we focused on their applications in cells, including the measurement of mechanical forces in the cell–ECM, which provides mechanical information for a better understanding of the interaction between cells and the matrix. Then, we summarized their applications in intercellular force measurement, including between single cells as well as collective cells, where these DNA tension probes can directly visualize natural intercellular interactions. Finally, we summarized the monitoring and regulatory role of this mechanical sensor on cellular mechanical function, indirectly regulating its function through changes in morphology. Based on the programmability, simplicity, and high flexibility of DNA, this modular tension

probe has gradually become an ideal tool for studying cellular mechanical information with ease and efficiency.

In addition to the commonly used DNA duplex and hairpin structures mentioned above, non-canonical nucleic acid structures such as G-quadruplex [67] and i-motif [68] have been shown to have better mechanical stability and relatively higher unfolding forces. Through rational design, they can be used to detect higher mechanical forces between cell surface receptor–ligand pairs in real time. Similarly, DNA origami, due to its rigid structure, can also provide higher mechanical forces. For example, DNA origami nanotubes described by Shrestha et al. [69] only unfold under a force of 40–50 pN. Therefore, selecting different types of nucleic acid structures as mechanical probes for different research objects can lead to faster and more accurate results. In addition, due to the limited detection rate and low throughput of microscopy technology, combining nucleic acid probes with microscopy technology and measuring the combination of force and fluorescence signals may directly measure the interaction between mechanical and chemical signals.

Although various DNA-based tension probes have been widely used in research on cells and non-cells, there are still some limitations. Firstly, there are challenges in the stability and specificity of membrane-anchored probes. Typically, probes use the hydrophobic interaction of cholesterol to rapidly and efficiently insert into the membrane, but their persistence on the membrane is challenging. They are usually endocytosed after staying on the membrane for 2–4 h [70]. Previous studies have shown that the hydrophobicity of lipid–DNA probes is related to the stability of membrane anchoring [71]. Therefore, the persistence of membrane anchoring can be adjusted by designing reasonable lipid-modified components or quantities. Due to the universality of cholesterol-anchoring membranes, some adapters or antibodies can be modified on the probes to achieve specificity. Meanwhile, DNA tension probes act on cells in a complex biological environment and may be affected by nucleases or proteases. Therefore, some DNA/peptide-related tension probes can be developed to enhance their resistance to nuclease degradation. Secondly, most existing studies focus on well-characterized cell adhesion molecules (integrin, cadherin, etc.). It is necessary to expand the research on other mechanical information caused by receptor–ligand interactions to better understand the mechanical properties of organisms. Finally, many cells may be subjected to mechanical forces in a high- or low-intensity range, and the measurement of larger mechanical forces may be limited by the weak hydrogen bond interaction between DNA probes, while the measurement of smaller mechanical forces may be difficult due to the design of low-threshold (<2 pN) probes. Therefore, the rational development of probes adapted to cell mechanics is still challenging.

In summary, due to the excellent and unique properties of DNA, it has become an ideal tool for mechanical sensors in cells, allowing us to have a clearer and more intuitive understanding of how mechanical information in organisms regulates signal transduction and mechanical function. Although there are still some challenges and limitations, we believe that the DNA tension probe platform will continue to improve and innovate, driving our understanding of various physiological and pathological processes.

**Author Contributions:** Conceptualization and writing—original draft preparation, X.S. and P.H.; writing—review and editing, X.S., P.H. and N.W. All authors have read and agreed to the published version of the manuscript.

**Funding:** This research was funded by the National Nature Science Foundation of China (No. 22174107, No. 31700861).

**Conflicts of Interest:** The authors declare no conflict of interest.

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
