# Peer review of "DNA-Based Mechanical Sensors for Cell Applications"

_chemistry, doi:10.3390/chemistry5030106_

Round 1

Reviewer 1 Report

Overall, this is an informative and well-written review article on using DNA and physical techniques to monitor cellular dynamics.  I have only two suggestions:

1. While well-written, many sections of the article are rather pedantic.  There is a lot of information provided that is really not necessary. For example, the Introduction describes actin/myosin which is not relevant to the topic of DNA-sensors.  I recommend a thorough revision for focus on DNA-mediated technologies.

2. The section on Conclusions is rather sparse.  I recommend adding a short paragraph highlighting real Future Directions in the field - would different types of nucleic acid provide better probes? would improvements in microscopy provide more sensitive measurements?  

Author Response

Response to Reviewer 1:

Comments: Overall, this is an informative and well-written review article on using DNA and physical techniques to monitor cellular dynamics.

Response: We thank the reviewer for his/her quite positive evaluation of our review.

Minor revisions:

  1. Comments: While well-written, many sections of the article are rather pedantic. There is a lot of information provided that is really not necessary. For example, the Introduction describes actin/myosin which is not relevant to the topic of DNA-sensors. Recommend a thorough revision for focus on DNA-mediated technologies.

Response: We thank the reviewer for this helpful comment. In Page 1, line 24-25, line 32-36, we have made appropriate deletions regarding the significance of studying mechanical forces in cells, and explained the importance of studying mechanical forces both inside and outside cells, as well as briefly describing the mechanical information related to human diseases.

In Page 2, line 58-65, we have added a paragraph focusing on the relevant applications and advantages of DNA nanostructure-mediated technologies. DNA has gradually emerged as a powerful tool for tension probes.

  1. Comments: The section on Conclusions is rather sparse. Recommend adding a short paragraph highlighting real Future Directions in the field - would different types of nucleic acid provide better probes? would improvements in microscopy provide more sensitive measurements?

Response: Thanks for pointing out this. In Page 10-11, line 428-440, we have added a paragraph on the advantages of using different types of DNA nanoscale probes, and emphasized the importance of selecting suitable probes according to the objects of study. Meanwhile, we expect that the probes can be combined with microscopy techniques, and the combination of force and fluorescence signal measurements will be able to directly measure the interaction between mechanical and chemical signals.

Reviewer 2 Report

This review explores DNA-based mechanical sensors and their various forms and developmental processes, such as DNA duplex, hairpin, and origami structures. These sensors offer technical support for the development of more effective probes. The review then focuses on the applications of these sensors in cells, particularly in measuring mechanical forces in the cell-extracellular matrix (ECM), which helps in understanding the interaction between cells and the matrix. It also discusses the applications of these sensors in measuring intercellular forces, both between single cells and collective cells, enabling the visualization of natural intercellular interactions. Additionally, the review summarizes how these mechanical sensors can monitor and regulate cellular mechanical function by indirectly influencing cellular morphology.

The author provides a detailed introduction to the different types of DNA used in DNA-based mechanical sensors and their applications across various fields. This encompasses numerous important references and research findings. However, there is relatively less emphasis on the design principles of core DNA-based mechanical sensors, particularly regarding how beginners can initiate the design process. For instance, there is a lack of specific explanations on determining the corresponding force values based on the number of base pairs. Enhancing this aspect of the review by including such introductions would significantly enhance its importance and value as a reference. By offering more guidance on design principles and force determination, this review could serve as a valuable resource for beginners to delve into and explore the design of DNA-based mechanical sensors.

The overall writing of this article is fluent, and the references are appropriately cited. If the aforementioned suggestions can be addressed and supplemented, it can be considered for publication.

Author Response

Response to Reviewer 2:

Comments: This review explores DNA-based mechanical sensors and their various forms and developmental processes, such as DNA duplex, hairpin, and origami structures. These sensors offer technical support for the development of more effective probes. The review then focuses on the applications of these sensors in cells, particularly in measuring mechanical forces in the cell-extracellular matrix (ECM), which helps in understanding the interaction between cells and the matrix. It also discusses the applications of these sensors in measuring intercellular forces, both between single cells and collective cells, enabling the visualization of natural intercellular interactions. Additionally, the review summarizes how these mechanical sensors can monitor and regulate cellular mechanical function by indirectly influencing cellular morphology.

Response: We thank the reviewer for his/her quite positive evaluation of our review.

Minor revisions:

  1. Comments: There is relatively less emphasis on the design principles of core DNA-based mechanical sensors, particularly regarding how beginners can initiate the design process. For instance, there is a lack of specific explanations on determining the corresponding force values based on the number of base pairs.

Response: In Page 2, line 78-87, we added a brief introduction to the design principles of DNA tension probes in the second part of the manuscript. Previous studies have used a combination of single-molecule force spectroscopy (SMFS) and computational modeling to obtain the mechanical information of DNA nanostructures.

In Page 2-3, line 96-102, due to the specific base-complementary pairing principle of DNA, the interaction strength between different base pairs varies. Therefore, probes with different tension thresholds can be designed using different types and numbers of bases, which enables better investigation of the mechanical properties of target molecules.